behaviour/ecology/environmental science

ecological trap, ghost fishing, marine debris, anthropogenic impacts, Paguroidea

**Author for correspondence:**
Atsushi Sogabe
e-mail: atsushi.sogabe@hirosaki-u.ac.jp

# Marine-dumped waste tyres cause the ghost fishing of hermit crabs

Atsushi Sogabe and Kiichi Takatsuji

Department of Biology, Hirosaki University, 3 Bunkyo-cho, Hirosaki, Aomori 036-8561, Japan

AS, 0000-0003-2915-8664

Poorly managed waste tyres pose serious environmental and health risks, ranging from air pollution caused by fire, leaching of heavy metals and outbreaks of mosquitos, to destruction of vegetation and coral reefs. We report a previously unrecognized ecological risk to marine organisms from waste tyres. Over 1 year, we made monthly counts of hermit crabs ($n = 1278$) invading and/or being trapped within six tyres anchored to the seabed at 8 m depth in Mutsu Bay, Japan. A complementary aquarium experiment in which hermit crabs were released into a tyre confirmed that they could not escape. We report marine-dumped waste tyres to ghost fish in a manner analogous to discarded fishing gear. Because hermit crabs play important roles in coastal food webs as both prey and scavengers, declines in their numbers as a consequence of this ghost fishing might affect coastal ecosystems.

## 1. Introduction

An estimated 29 million metric tons of waste tyres are generated annually by 45 countries that use 83.5% of vehicles in the world [1]. Although most (*ca* 90%) waste tyres are repurposed as sources of energy (i.e. tyre-derived fuel) or materials (e.g. reclaimed rubber), a significant number of tyres go to landfill, are stockpiled or are illegally dumped [1]. These discarded tyres pose serious environmental and health risks, from air pollution following fire, and soil and water-heavy metal pollution, to outbreaks of mosquitos that mediate various infectious diseases [2].

Waste tyres can also enter the marine environment through various routes, such as missing dock/boat fenders, poorly engineered artificial tyre reefs and illegal dumping [3]. As known in the notorious case of Osborne Tyre Reef project in Florida [4], tyres in the marine environment can destroy salt marsh vegetation and coral reefs when they drift ashore, move in currents or by wave action [4,5]. We report an overlooked ecological risk of marine-dumped waste tyres to marine organisms.

This research was initiated after observing large numbers of gastropod shells within a tyre on the seabed at 8 m depth during a

**Figure 1.** (*a,b*) Tyre on the seabed at 8 m depth, Mutsu Bay, June 2012, with numerous empty gastropod shells, some with hermit crabs. (*c*) Monthly catch of hermit crab species (*bars*) and cumulative catch (*line*) from October 2015 to September 2016.

survey in Mutsu Bay, Japan, in June of 2012. Some of these shells were occupied by hermit crabs (figure 1*a,b*). We hypothesized that these crabs could not escape the tyre once inside because of the concave inner tyre wall, and that they would die there. An opportunity later presented itself to conduct field and aquarium experiments to verify this hypothesis, setting tyres on the seabed for 1 year to monitor numbers of hermit crabs entering them, and releasing hermit crabs inside a tyre in an aquarium to ascertain their escape ability.

Hermit crabs are crustaceans belonging to the Superfamily Paguroidea, most of which inhabit empty gastropod shells to protect non-calcified 'soft' abdomens. More than 800 described species were recognized from the world's ocean, from intertidal area to deep seas, exceptionally with some terrestrial or freshwater species [6]. Hermit crabs occupy an important position in the food web of coastal ecosystems, not only as food for other animals such as fish, but also as scavengers that feed on the carcasses of animals and plants [7]. In addition, hermit crabs function as ecosystem engineers, providing a habitat for other organisms by re-using gastropod shells, which are usually buried on the sea floor [8]. Therefore, a decrease in the diversity and biomass of hermit crabs could have a significant impact on coastal ecosystems.

# 2. Material and methods

## 2.1. Field experiment

We monitored waste tyres set in Mutsu Bay, Japan (40°54′14.2″ N, 140°51′20.5″ E). Six passenger car tyres (diameter 664.4–688 mm, width 215–235 mm) were set on the seabed at about 8 m depth with their sides

up in June 2014; tyres were partially buried into the seabed (mainly pebbles, sand and mud, with small stones and shells) to a depth approximately one-quarter of their width. To prevent tyres from being dislodged by currents, they were further affixed to the seabed using steel tent pegs (10 mm diameter × 300 mm length). In order to mimic the condition of tyres that have been submerged on the seafloor for a long period of time, the tyres were set 18 months prior to the start of the monthly observation, allowing the growth of sea algae and attachment of sessile invertebrates on the surface of the tyre. Hermit crabs and other large benthos inside the tyre were removed once a month until the observation began.

All hermit crabs in each tyre were collected monthly from October 2015 to September 2016. Crabs were photographed (with a scale), using a binocular microscope for species identification and to measure body size. Shield length (the calcified anterior portion of the cephalothorax) was taken as an index of body size from digital images using the ImageJ program (National Institute of Health, USA). When the shell was held with the shell mouth facing upward, the hermit crab inside the shell exposed its upper body, making it easy to photograph. Juveniles with undeveloped taxonomic traits were classified 'unidentified'. All crabs were released at least 50 m from where they were collected after measurement. Although the possibility of released hermit crabs re-entering the tyre cannot be ruled out, recapture is unlikely to have much effect on the interpretation of the results because hermit crabs do not show homing behaviour [9], there are many potential habitats around the release site, and the biomass of hermit crabs is very high in the study area.

## 2.2. Aquarium experiment

To determine if hermit crabs could escape from the inner area of a horizontally set tyre, we placed a tyre used in the field experiment within the centre of a round 350 l polyethylene tank (98 cm diameter × 46 cm high) and buried in sand to approximately one-quarter of its width. The tank was continuously filled with fresh seawater. The overflowing seawater was drained and not recirculated. The experiment ran from 31 October to 14 December 2016.

The two most common hermit crab species (*Pagurus minutus* and *Paguristes ortmanni*) found in tyres in the field experiment were used in the laboratory experiment. In each of six replicate experiments for each species, 10 of each hermit crab species were placed outside (hereafter referred to as *intruder*) or inside (hereafter *escaper*) a tyre at 15.00. After 18 h (at 09.00), the position of each individual was recorded (whether inside or outside the tyre). The *intruder* and *escaper* experiments were conducted separately. For each species, the shield length of intruder and escaper crabs did not differ. *Pagurus minutus* intruder mean ± s.d. = 4.84 ± 1.00 mm ($n = 60$); escaper: 4.87 ± 0.97 mm ($n = 59$; one individual which could not be measured was excluded), Student's $t$-test, $t = 0.14$, $p = 0.89$. *Paguristes ortmanni* intruder mean ± s.d. = 5.78 ± 1.14 mm ($n = 60$); escaper 5.78 ± 1.19 mm ($n = 60$), $t = 0.01$, $p = 0.99$). Each hermit crab was used only once for the experiment and was subsequently released near their capture site. We conducted field and captive experiments following the Hirosaki University guidelines for the ethical treatment of animals.

All statistical analyses were performed using R v. 4.0.3 [10]. Student's $t$-test was used to confirm there was no size difference between the *intruder* and *escaper*. Fisher's exact test was used to verify whether escape from the tyre was more difficult than entry into the tyre.

## 3. Results and discussion

Intentionally or unintentionally abandoned fishing gear (e.g. trammel and gill nets, cage traps and crab pots) can continue to catch and entangle aquatic animals without human control. This 'ghost fishing' phenomenon kills a wide variety of aquatic animals, from marine mammals, seabirds, sea turtles, fishes to molluscs [11,12]. We reveal ocean-dumped tyres to also 'ghost fish' hermit crabs in an analogous process.

Over 1 year, 1278 hermit crabs were trapped within six horizontally set tyres (figure 1*c*), averaging 0.58 hermit crabs per tyre per day. Most specimens were referable to *P. ortmanni* ($n = 604$, 47.3%) and *P. minutus* ($n = 592$, 46.3%), with *P. lanuginosus* ($n = 54$, 4.2%), *P. middendorffii* ($n = 10$, 0.8%), *P. nigrofascia* ($n = 3$, 0.2%), *Areopaguristes japonicus* ($n = 3$, 0.2%), *P. quinquelineatus* ($n = 1$, 0.1%), and 11 unidentified juvenile specimens, being of secondary importance. This may reflect differences in microhabitat among species, with *P. ortmanni* and *P. minutus* preferring muddy or sandy intertidal to subtidal substrata [13,14]. The wide range in body size of hermit crabs entering tyres indicates that size did not limit tyre entry (mean shield length *P. ortmanni* (± s.d.) = 5.70 ± 1.84 mm, range 0.7–15.3 mm, $n = 604$; *P. minutus* 2.30 ± 1.12 mm, range 0.9–10.0 mm, $n = 590$ (excluding two individuals for which measurements were not possible);

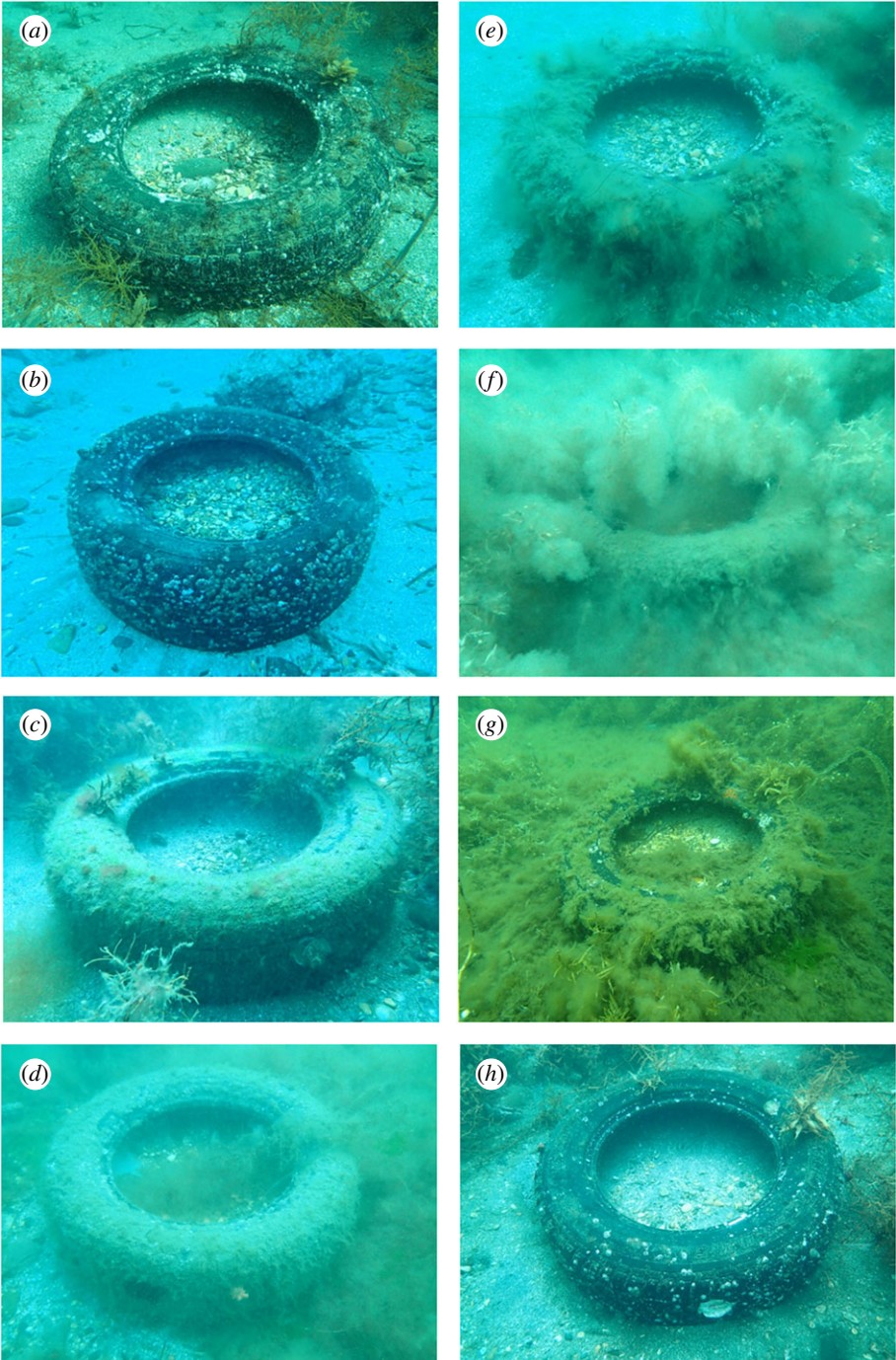

**Figure 2.** Seasonal changes in algal growth on and around the experimental tyre. Photographed in (*a*) October 2015, (*b*) January 2016, (*c*) March 2016, (*d*) April 2016, (*e*) May 2016, (*f*) June 2016, (*g*) July 2016 and (*h*) August 2016.

*P. lanuginosus*, $6.59 \pm 0.94$ mm, range: 4.5–9.3 mm, $n = 54$; others: $2.27 \pm 1.61$ mm, range 0.7–6.3 mm, $n = 28$; see also electronic supplementary material, figures S1 and S2).

Numbers of hermit crabs entering tyres tended to increase during winter and decrease from spring to early summer, peaking at 246 individuals in March and troughing at 19 individuals in June (figure 1*b*). By species, *P. minutus*, *P. ortmanni* and *P. langinosus* were found in the tyres in the highest numbers in January, March and January, respectively, while the lowest number of captures for any species was recorded in June. Because the main local hermit crab predators, fat greenling *Hexagrammos otakii* [15] and Schlegel's black rockfish *Sebastes schlegelii*, overwinter in deeper waters, reduced predation risk in winter may increase hermit crab activity and contribute to their higher numbers being found in tyres. Alternatively, decreased catch from spring to early summer may be related to rapid spring seaweed growth on and around tyres, which hermit crabs within tyres might exploit to escape, leading to

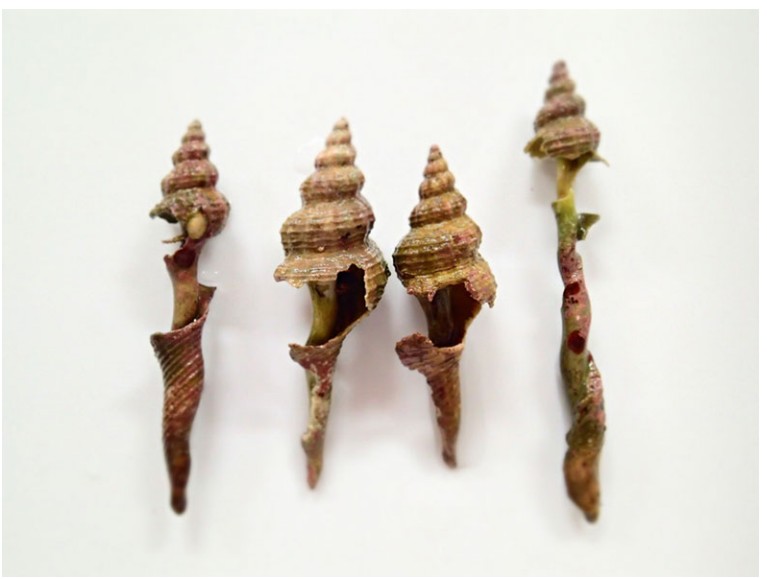

**Figure 3.** Hermit crabs living in gastropod shells with damaged spires collected from the tyre found in June 2012.

underestimates of their numbers invading tyres (figure 2). Additionally, when seaweed dies in early summer, the adjacent seabed becomes anaerobic, reducing hermit crab activity.

Aquarium experiments confirmed that hermit crabs cannot escape from tyres within a short period of time, with no *P. minutus* ($n = 60$) or *P. ortmanni* ($n = 60$) escaping a tyre within an 18 h period. Six *P. minutus* (1, 2 and 3 individuals) and 13 *P. ortmanni* (3, 3, 3 and 4 individuals) entered tyres in replicate trials for each species. Due to the small sample size, we were not able to statistically show that escape from the tyre was less likely to occur than entry into the tyre (Fisher's exact test, *P. minutus*: $p = 0.18$, *P. ortmanni*: $p = 0.06$). Although hermit crabs could escape from a tyre given sufficient time, this is unlikely because the inner tyre surface is smooth, and without encrusting organisms, there are no footholds to assist with climbing even after 2 years in the field when the outer tyre surface is covered with epifauna (algae, barnacles, oysters and other sessile organisms; figure 2).

The length of time that fishing gear can ghost fish ranges from a few months to 3 years, depending on gear type, what it is made of, and the environment into which it was discarded [16,17]. Because of their simple structure, temporal persistence and robustness, tyres may ghost fish hermit crabs for considerably longer. Ecological trap occurs when animals were mistakenly attracted to the cue uninformative of habitat quality and select a habitat where their fitness is lower than in other available options [18]. At this time, it is unclear whether a marine-dumped waste tyre is an ecological trap for hermit crabs, as there is no solid evidence that hermit crabs were attracted to tyres and that hermit crabs trapped in tyres lowered their fitness. However, circumstantial evidence suggests that intense competition for food and gastropod shells, which are essential for survival and growth of hermit crabs, is likely to have reduced the fitness of them inside the tyre. This was inferred from the fact that shells occupied by hermit crabs inside the tyre we found in June 2012 were heavily damaged, suggesting that competition for lodgings and cannibalism may be occurring (figure 3). It is not known how long hermit crabs can survive starvation, but if a tyre can ghost fish beyond that period then mortality is likely. Thus, in bays and lagoons where currents and wave action are weakened and the seabed is covered with sand or mud, tyres that are unlikely to move may pose serious threats to hermit crab survival.

The negative impact of tyres on hermit crabs was unexpected, and the effects of ghost fishing on their populations, and cascading effects these may have on coastal communities and ecosystems, are unknown. Because hermit crabs are important as both prey and scavengers [7,19] and ecosystem engineers [8], the extent of the ghost fishing effect that tyres may have on them and other benthos warrants further study.

Data accessibility. The data are provided in the electronic supplementary material [20].

Authors' contributions. A.S. designed the study. A.S and K.T. conducted field and captive experiments and analysed data. A.S. produced the figures and first draft of the manuscript.

Competing interests. We declare that we have no competing interests.

Funding. The research was supported by the Nippon Life Insurance Foundation to A.S. (grant no. 2016-13).

Acknowledgements. We thank Masahiko Washio, Hirokazu Abe and Takehiro Endo for assistance with fieldwork, and Steve O'Shea, PhD, from Edanz Group (https://en-author-services.edanz.com/ac) for editing a draft of the manuscript. This study was supported by the MEXT programme in Research Center for Marine Biology Asamushi, Tohoku University.

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
