## [Peer Review File · Royal Society Open Science]

Review History

RSOS-210166.R0 (Original submission)

Review form: Reviewer 1

Is the manuscript scientifically sound in its present form?

No

Are the interpretations and conclusions justified by the results?

Yes

Is the language acceptable?

Yes

Do you have any ethical concerns with this paper?

No

Have you any concerns about statistical analyses in this paper?

Yes

Recommendation?

Major revision is needed (please make suggestions in comments)

Comments to the Author(s)

The authors raise an interesting question in exploring the consequences of a common form of marine pollution on an intertidal species group. The authors provide a simple but straightforward presentation of this seemingly unknown phenomena of hermit crabs getting trapped in tires. There are only basic summary statistics reported, which may be fine if this is treated as a natural history note, but may not be rigorous enough if this is to be treated as an empirical paper. I also have questions about the methods that need some clarification, and think both the introduction and discussion could do with greater context. Specifically, the paper would be much improved with reference to hermit crab ecology or species' prevalence to help readers understand the scope of the problem, and with also placing the study within the larger literature on ecological traps. It appears the authors are proposing to have found a new form of ecological trap, but do not call it so by name, and therefore miss out on the potential links with that entire field.

My detailed comments can be found below.

Introduction:

The authors clearly explain why tires can pose an environmental hazard. However, they do not introduce hermit crabs at all, apart from when they mention their hypothesis that tires could collect crabs. Even just a small paragraph about the prevalence of hermit crabs in marine environments, or anything about their biology could really help contextualize why this source of marine pollution would potentially have a large impact if it truly traps individuals.

Methods

Line 53- What happened to the tires between June 2014 and October 2015 if data collection did not begin until October 2015? Was there a reason for this long delay, such as making the tires more attractive or allowing them to form some algal growth?

Line 57- I believe the authors meant "held".

Line 59- Given that hermit crabs can travel >50m in a year long period, did the authors do anything to account for these potentially repeated measurements? Or do the authors have any evidence they can provide to suggest that re-trapping was unlikely? Some additional detail or justification for the 50m distance would be helpful.

Line 66-67: There seems to be a discrepancy with the timelines the authors present here. If the six tires were in the field until at least September 2016, how is it possible that a tire used from the field experiment from October to December 2016?

Line 69-70: Was each hermit crab used only once or were any used in multiple replicates? Were hermit crabs marked at all to distinguish between the intruders and escapers?

There is no mention in the methods about what statistics are to be performed and what data will be compared to determine their conclusions. For instance, there are no analyses on seasonal or species level differences.

Results/Discussion

Line 119-121: This might be a good place to bring in a short discussion of how these tires could represent an ecological trap for hermit crabs. There is a rich literature there, which could help

give the reader greater insight into the seriousness of the problem and how it relates to other impacts from man-made objects and pollution.

Figure 1: This figure is very helpful and clear.

There was no mention of ethical approval for the use of animal subjects. Given that invertebrates were used, it is not clear what approval was needed from the home institution of the authors.

Supplementary materials:

While the data are clear and readily accessible, there was no code included with this article. It is not clear how the summary statistics were calculated, but they are simple enough that they could have been conducted directly in the Excel spreadsheets. Some clarity here would be helpful from the authors.

Decision letter (RSOS-210166.R0)

Dear Dr Sogabe

The Editors assigned to your paper RSOS-210166 "Marine-dumped waste tyres cause the ghost fishing of hermit crabs" have now received comments from reviewers and would like you to revise the paper in accordance with the reviewer comments and any comments from the Editors. Please note this decision does not guarantee eventual acceptance.

Please submit your revised manuscript and required files (see below) no later than 21 days from today's (ie 11-Aug-2021) date. Note: the ScholarOne system will 'lock' if submission of the revision is attempted 21 or more days after the deadline. If you do not think you will be able to meet this deadline please contact the editorial office immediately.

on behalf of Dr Punidan Jeyasingh (Associate Editor) and Pete Smith (Subject Editor)
openscience@royalsociety.org

Associate Editor Comments to Author (Dr Punidan Jeyasingh):

This is an interesting manuscript, representing significant man-hours of work. I had a very difficult time securing expert reviews for this manuscript. It was assessed by an expert who was enthusiastic about the study, but makes a number of points that need to be addressed. I felt the review was fair, clear, and constructive. With much gratitude to the expert reviewer, I invite the authors to make these revisions.

Reviewer comments to Author:

Reviewer: 1

Comments to the Author(s)

The authors raise an interesting question in exploring the consequences of a common form of marine pollution on an intertidal species group. The authors provide a simple but straightforward presentation of this seemingly unknown phenomena of hermit crabs getting trapped in tires. There are only basic summary statistics reported, which may be fine if this is treated as a natural history note, but may not be rigorous enough if this is to be treated as an empirical paper. I also have questions about the methods that need some clarification, and think both the introduction and discussion could do with greater context. Specifically, the paper would be much improved with reference to hermit crab ecology or species' prevalence to help readers understand the scope of the problem, and with also placing the study within the larger literature on ecological traps. It appears the authors are proposing to have found a new form of ecological trap, but do not call it so by name, and therefore miss out on the potential links with that entire field.

My detailed comments can be found below.

Introduction:

The authors clearly explain why tires can pose an environmental hazard. However, they do not introduce hermit crabs at all, apart from when they mention their hypothesis that tires could collect crabs. Even just a small paragraph about the prevalence of hermit crabs in marine environments, or anything about their biology could really help contextualize why this source of marine pollution would potentially have a large impact if it truly traps individuals.

Methods

Line 53- What happened to the tires between June 2014 and October 2015 if data collection did not begin until October 2015? Was there a reason for this long delay, such as making the tires more attractive or allowing them to form some algal growth?

Line 57- I believe the authors meant "held".

Line 59- Given that hermit crabs can travel >50m in a year long period, did the authors do anything to account for these potentially repeated measurements? Or do the authors have any evidence they can provide to suggest that re-trapping was unlikely? Some additional detail or justification for the 50m distance would be helpful.

Line 66-67: There seems to be a discrepancy with the timelines the authors present here. If the six tires were in the field until at least September 2016, how is it possible that a tire used from the field experiment from October to December 2016?

Line 69-70: Was each hermit crab used only once or were any used in multiple replicates? Were hermit crabs marked at all to distinguish between the intruders and escapers?

There is no mention in the methods about what statistics are to be performed and what data will be compared to determine their conclusions. For instance, there are no analyses on seasonal or species level differences.

Results/Discussion

Line 119-121: This might be a good place to bring in a short discussion of how these tires could represent an ecological trap for hermit crabs. There is a rich literature there, which could help give the reader greater insight into the seriousness of the problem and how it relates to other impacts from man-made objects and pollution.

Figure 1: This figure is very helpful and clear.

There was no mention of ethical approval for the use of animal subjects. Given that invertebrates were used, it is not clear what approval was needed from the home institution of the authors.

Supplementary materials:

While the data are clear and readily accessible, there was no code included with this article. It is not clear how the summary statistics were calculated, but they are simple enough that they could have been conducted directly in the Excel spreadsheets. Some clarity here would be helpful from the authors.

===PREPARING YOUR MANUSCRIPT===

Your revised paper should include the changes requested by the referees and Editors of your manuscript. You should provide two versions of this manuscript and both versions must be provided in an editable format:
 one version identifying all the changes that have been made (for instance, in coloured highlight, in bold text, or tracked changes);
 a 'clean' version of the new manuscript that incorporates the changes made, but does not highlight them. This version will be used for typesetting if your manuscript is accepted.

===PREPARING YOUR REVISION IN SCHOLARONE===

Author's Response to Decision Letter for (RSOS-210166.R0)

See Appendix A.

Decision letter (RSOS-210166.R1)

Dear Dr Sogabe,

It is a pleasure to accept your manuscript entitled "Marine-dumped waste tyres cause the ghost fishing of hermit crabs" in its current form for publication in Royal Society Open Science. The comments of the reviewer(s) who reviewed your manuscript are included at the foot of this letter.

Kind regards,
Royal Society Open Science Editorial Office
Royal Society Open Science

on behalf of Dr Punidan Jeyasingh (Associate Editor) and Pete Smith (Subject Editor)
openscience@royalsociety.org

Associate Editor Comments to Author (Dr Punidan Jeyasingh):

Associate Editor

Comments to the Author:

I thank the authors for carefully addressing reviewer comments. I am happy to recommend this manuscript for publication.

Appendix A

Reply to the reviewer's comments

Reviewer #1's Comments to the Author(s):

The authors raise an interesting question in exploring the consequences of a common form of marine pollution on an intertidal species group. The authors provide a simple but straightforward presentation of this seemingly unknown phenomena of hermit crabs getting trapped in tyres. There are only basic summary statistics reported, which may be fine if this is treated as a natural history note, but may not be rigorous enough if this is to be treated as an empirical paper. I also have questions about the methods that need some clarification, and think both the introduction and discussion could do with greater context. Specifically, the paper would be much improved with reference to hermit crab ecology or species' prevalence to help readers understand the scope of the problem, and with also placing the study within the larger literature on ecological traps. It appears the authors are proposing to have found a new form of ecological trap, but do not call it so by name, and therefore miss out on the potential links with that entire field.

Reply: Thank you for all your comments and suggestions. Following the comments from you, we substantially revised the Introduction and Discussion section to add information on ecological traps and ecology of hermit crabs. We also made some modification to improve readability of the paper in the M & M and R & D section. Please see our response to each of the specific comment as below.

1. The authors clearly explain why tyres can pose an environmental hazard. However, they do not introduce hermit crabs at all, apart from when they mention their hypothesis that tyres could collect crabs. Even just a small paragraph about the prevalence of hermit crabs in marine environments, or anything about their biology could really help contextualize why this source of marine pollution would potentially have a large impact if it truly traps individuals.

Reply: Thanks for your suggestion. I added a paragraph describing the biology of hermit crabs, especially in their ecological roles in coastal ecosystem, in the Introduction section (L.44 -52 in the revised version).

2. Line 53- What happened to the tyres between June 2014 and October 2015 if data collection did not begin until October 2015? Was there a reason for this long delay, such as making the tyres more attractive or allowing them to form some algal growth?

Reply: If a tyre loses its ability to ghost fish in a short period of time due to the attachment of sessile organisms (e.g., s sea weeds and barnacles), the effect of a tyre discarded in the ocean would not be a big deal. Our concern is whether ocean-dumped

tyres will maintain their ability to ghost fish over the long period of time. In order to replicate the conditions of tyres submerged in the ocean for a long period of time, we set the tyres more than a year before the start of the experiment. In the revised paper, the intent of the experimental procedure was clearly stated in the Material & Methods section (L. 61-64 in the revised version).

3. Line 57- I believe the authors meant “held” .

Reply: We replaced “hold” with “held.”

4. Line 59- Given that hermit crabs can travel >50m in a year long period, did the authors do anything to account for these potentially repeated measurements? Or do the authors have any evidence they can provide to suggest that re-trapping was unlikely? Some additional detail or justification for the 50m distance would be helpful.

Reply: As you mentioned, this method does not completely eliminate the possibility that the same individual repeatedly invaded the tyre. However, since hermit crabs do not show homing behavior (Vannini & Cannicci 1995), and there are potential habitats everywhere in the study area, the proportion of hermit crabs that return to the tyre is considered low. Also, since the biomass of hermit crabs at the study site is very large, recapture of the same individual would not affect the interpretation of the results so much. In the revised manuscript, we mentioned the possibility that released hermit crabs could be captured by tyres again (L. 72-75 in the revised version).

Vannini, M. & Cannicci, S. (1995) Homing behaviour and possible cognitive maps in crustacean decapods. *J. Exp. Mar. Biol. Ecol.* 193: 67-91.

5. Line 66-67: There seems to be a discrepancy with the timelines the authors present here. If the six tyres were in the field until at least September 2016, how is it possible that a tyre used from the field experiment from October to December 2016?

Reply: The field experiment ended on 16 September 2016, so there is no contradiction if the aquarium experiment started from 31 October 2016 using the tyre that was once set in the field.

6. Line 69-70: Was each hermit crab used only once or were any used in multiple replicates? Were hermit crabs marked at all to distinguish between the intruders and escapers?

Reply: The explanation was insufficient. Each hermit crab was used only once for the experiment. Since the experiments to verify escape from and entry into the tyre were

conducted separately, we did not need to mark them for group identification (i.e., intruder and escaper). The revised manuscript clarified these points (L.87-88 in the revised version).

7. There is no mention in the methods about what statistics are to be performed and what data will be compared to determine their conclusions. For instance, there are no analyses on seasonal or species level differences.

Reply: We added explanation for the statistical analysis methods in the Material & Method section (L. 95-97 in the revised version). Additional analyses were conducted to statistically show that escape from the tyre was less likely to occur than entry into the tyre, but the results were not significant due to the small sample size (L. 132-134 in the revised version). We agree with the reviewer's point about the need for statistical analysis of field experiment data, but our data are insufficient to perform the required statistical analysis. Therefore, we only report the seasonal patterns and species-specific trends of hermit crabs captured in tyres in a descriptive manner (L. 119-121 in the revised version).

8. Line 119-121: This might be a good place to bring in a short discussion of how these tyres could represent an ecological trap for hermit crabs. There is a rich literature there, which could help give the reader greater insight into the seriousness of the problem and how it relates to other impacts from man-made objects and pollution.

Reply: Following the suggestion, we introduced the concept of ecological trap here and mentioned the possibility of tyres being an ecological trap for hermit crabs (L. 141-149). At this time, it is unclear whether tyres are an ecological trap for hermit crabs, as there is no solid evidence that tyres reduce the fitness of hermit crabs, but we have provided indirect evidence of increased competition between individuals caused by a paucity of food and gastropod shells. In addition, we added the term 'ecological trap' to the keywords to make it easier for potential readers to find this paper.

9. Figure 1: This figure is very helpful and clear.

Reply: Thanks.

10. There was no mention of ethical approval for the use of animal subjects. Given that invertebrates were used, it is not clear what approval was needed from the home institution of the authors.

Reply: We added ethical statement in the Material & methods section (L. 93-94 in the revised version).

11. While the data are clear and readily accessible, there was no code included with this article. It is not clear how the summary statistics were calculated, but they are simple enough that they could have been conducted directly in the Excel spreadsheets. Some clarity here would be helpful from the authors.

Reply: We did not use code for statistical analyses in this article. The open data accompanying this paper has been prepared with reference to that of other papers published in the Royal Society Open Science.